# 5-Iodo-4-thio-2′-Deoxyuridine as a Sensitizer of X-ray Induced Cancer Cell Killing

**DOI:** 10.3390/ijms20061308

**Published:** 2019-03-15

**Authors:** Samanta Makurat, Paulina Spisz, Witold Kozak, Janusz Rak, Magdalena Zdrowowicz

**Affiliations:** Faculty of Chemistry, University of Gdańsk, Wita Stwosza 63, 80-308 Gdańsk, Poland; samanta.makurat@ug.edu.pl (S.M.); paulina.rewers@phdstud.ug.edu.pl (P.S.); davelombardo@wp.pl (W.K.); janusz.rak@ug.edu.pl (J.R.)

**Keywords:** radiosensitizer, radiotherapy, X-ray, modified nucleosides

## Abstract

Nucleosides, especially pyrimidines modified in the C5-position, can act as radiosensitizers via a mechanism that involves their enzymatic triphosphorylation, incorporation into DNA, and a subsequent dissociative electron attachment (DEA) process. In this paper, we report 5-iodo-4-thio-2′-deoxyuridine (ISdU) as a compound that can effectively lead to ionizing radiation (IR)-induced cellular death, which is proven by a clonogenic assay. The test revealed that the survival of cells, pre-treated with 10 or 100 µM solution of ISdU and exposed to 0.5 Gy of IR, was reduced from 78.4% (for non-treated culture) to 67.7% and to 59.8%, respectively. For a somewhat higher dose of 1 Gy, the surviving fraction was reduced from 68.2% to 54.9% and to 40.8% for incubation with 10 or 100 µM ISdU, respectively. The cytometric analysis of histone H2A.X phosphorylation showed that the radiosensitizing effect of ISdU was associated, at least in part, with the formation of double-strand breaks. Moreover, the cytotoxic test against the MCF-7 breast cancer cell line and human dermal fibroblasts (HDFa line) confirmed low cytotoxic activity of ISdU. Based on the results of steady state radiolysis of ISdU with a dose of 140 Gy and quantum chemical calculations explaining the origin of the MS detected radioproducts, the molecular mechanism of sensitization by ISdU was proposed. In conclusion, we found ISdU to be a potential radiosensitizer that could improve anticancer radiotherapy.

## 1. Introduction

Modified nucleosides are well suited for radiation-induced cell killing because of their unique features. They can be incorporated (at least some of them) into native DNA without affecting its function. Moreover, if modified appropriately, they may produce lethal effects as a result of interactions between radiation and the labeled DNA. For these reasons, the modified nucleosides are considered as radio- and photosensitizers with potential employment in radiotherapy [1,2,3].

Some of the most widely studied in this group of compounds are halogen derivatives [4,5,6,7]. Numerous studies, both in vitro and in vivo, have proven that 5-bromo-2′-deoxyuridine (BrdU) and 5-iodo-2′-deoxyuridine (IdU) are effectively incorporated (as triphosphates) into the DNA chain by DNA polymerases [8]. An in vitro treatment with BrdU or IdU followed by exposure to ionizing or UV radiation leads to an increase of activity in metabolic pathways signaling DNA damage [9,10]. In both cases (radiolysis and photolysis), the degradation process produces reactive radicals, which can cause serious damage to the biopolymer, such as single- and double-strand breaks as well as extremely cytotoxic crosslinks. It is worth mentioning that, regardless of the dose of sparsely ionizing radiation (IR), around 60% of the single strand breaks (SSB)s are composed of one strand break while ca. 40% are comprised of two or more lesions [11]. The latter, so called cluster (or complex) lesions, seem to be the most important biologically [12]. Indeed, the repair of cluster damage is impaired as compared to the isolated breaks or abasic sites. As a consequence, the lifetimes of complex lesions are significantly longer than those of isolated ones and they may survive till the S-phase of the cell cycle [13]. Moreover, attempted post-irradiation repair may transform some complex non- double-strand break (DSB) clusters into DSBs [14]. Under hypoxia, the BrdU or IdU labeled DNA is sensitive to solvated electrons which are non-consequential toward the native biopolymer [15]. Since the ionizing radiation (IR)-induced solvated electrons are as abundant as hydroxyl radicals—the main source of indirect DNA damage [12]—the probability of cluster damage formation should significantly increase due to the incorporated halogen derivatives of uridine.

An interesting group of modified nucleosides are derivatives of 4-thio-2′-deoxyuridine. The presence of the sulfur atom causes a bathochromic shift of the absorption maximum of 2′-deoxyuridine to the UVA region (approximately to 340 nm) [16]. In contrast to UVC or UVB radiation, the less energetic UVA radiation is absorbed very poorly by native DNA and therefore causes relatively little DNA damage. However, DNA labeled with a modified nucleoside, being a UVA chromophore, can be damaged by normally harmless UVA radiation. Indeed, 5-iodo-4-thio-2′-deoxyuridine (ISdU) and 5-bromo-4-thio-2′-deoxyuridine (BrSdU) are confirmed photosensitizers of DNA damage [17]. It has been proven that ISdU and BrSdU are incorporated into the DNA of cultured human and mouse cells and sensitize killing with low doses of UVA radiation by generation of interstrand crosslinks, DNA–protein crosslinks, DNA strand breaks, nucleobase damage, and pyrimidine–pyrimidone (6-4) photoproducts [18,19].

In the present paper, we demonstrate that ISdU is also an effective sensitizer to ionizing radiation (IR)-induced cellular death. The potential usage of ISdU as a selective radiosensitizer is related to the involvement of solvated electrons in the DNA damage process. These species are one of the major products of water radiolysis under hypoxic conditions (characteristic for solid tumors) [20]. We performed a clonogenic assay to prove that the treatment of MCF-7 breast cancer cells with ISdU, demonstrating very low cytotoxicity in the MTT test at concentrations typical for cell labeling, results in a decrease of cell survival after irradiation with IR doses as low as 0.5 and 1.0 Gy. A histone H2A.X phosphorylation assay was used to determine the relationship between γH2A.X formation and radiosensitivity of ISdU-labeled MCF-7 cells. Moreover, a cytometric assay which allows the evaluation of change in mitochondrial potential (early apoptosis and cellular stress), phosphatidylserine expression on the cell surface (late apoptosis), and membrane permeabilization (cell death) were performed. In order to elucidate the molecular mechanism behind the increased radiosensitivity of cells treated with the sensitizer, we performed radiation chemistry experiments for aqueous solutions of ISdU. Thus, we carried out steady state radiolysis of the deoxygenated solution of ISdU with an ^●^OH radical scavenger and the mechanisms responsible for the formation of the observed radioproducts were proposed with the use of quantum chemical calculations. Our results suggest that the combined use of ISdU and X-rays opens a promising route for effective killing of cancer cells.

## 2. Results and Discussion

Halogenated nucleosides are usually both photo- and radiosensitive [1]. Previously (in 1960), Greer demonstrated [21] that BrdU efficiently photosensitizes *Escherichia coli* to UV light. Concurrently, the Szybalski group observed increased radiosensitivity of human cell lines labeled with this derivative [22]. As was mentioned above, ISdU has well documented photosensitizing properties and, when incorporated into cellular DNA, makes the labeled cells sensitive to UVA photons [17,18,19]. Reasoning by analogy, one can thus presume that ISdU, besides photosensitizing properties, should have also radiosensitizing features. Surprisingly, the nucleoside, which is accepted by human nucleoside kinases and DNA polymerases [18], has never been studied (neither in vitro nor in vivo) for its radiosensitizing properties. In order to fill this gap, we first examined the propensity of ISdU to be damaged by solvated electrons, which are one of the main products of water radiolysis under hypoxia. It is worth mentioning that the nucleoside derivatives, which are not damaged by solvated electrons, are not able to sensitize the hypoxic cells to IR [1]. Then, we assessed cell reproductive death after treatment with ISdU and a specific dose of IR, with the method of choice for evaluating the possibility of neoplasm reoccurrence—the clonogenic assay [23]. Finally, the cytotoxicity of ISdU was examined, since only these derivatives, which are characterized by low toxicity, can be used as radiosensitizers.

### 2.1. Radiolysis

In order to investigate the susceptibility of ISdU to solvated electron-induced degradation, steady state radiolysis was carried out in a phosphate buffer solution in the presence of an ^●^OH radical scavenger. In addition, prior to irradiation with a dose of 140 Gy, the studied mixture was deoxygenated to avoid scavenging of electrons by oxygen and to create conditions similar to the hypoxia in a tumor cell. After irradiation, the solution was analyzed by the HPLC and LC–MS methods. The chromatogram in Figure 1 shows that radiolysis of the ISdU solution gave five main radioproducts (for their chemical structures see Scheme 1). Their identification was carried out using LC–MS/MS (for fragmentation spectra and ion identities see Appendix A). One of the main products of radiolysis was SdU (Scheme 1, path A), which indicates that dissociative electron attachment (DEA) to ISdU lead to the SdU^●^ radical and iodide anion. Two further products, eluting with retention times of 4.62 min and 8.21 min, respectively, were IdU and ISOdU (Scheme 1, paths B and C). The last two signals (see Figure 1) corresponded to dimers in which the ISdU monomers were connected with a disulfide bond (see the ISdU–SdU and (ISdU)_2_ structures in Scheme 1, paths D and E). The formation of dimers indicates engagement of the S-centered radicals in pathways of ISdU degradation by IR.

Steady state radiolysis coupled with quantitative HPLC analysis enabled us to estimate the degree of ISdU degradation. Comparison of the area of signals, corresponding to ISdU before and after irradiation, showed that the extent of damage induced by X-rays (140 Gy) was equal to 27.31 ± 2.43%. Under the same conditions, it was worth noting that we measured the extent of BrdU decay as equal to 19.31 ± 0.86%. Thus, the obtained data suggest the superiority of ISdU over BrdU.

### 2.2. Computational

In order to simplify the calculations, we substituted the 2′-deoxyribose residue, which is not involved in the IR-induced damage, with a methyl group and dubbed it ISU (see Scheme 2). We calculated the mechanisms leading to the expected DEA product (denoted SU in Scheme 2), two types of dimers ((ISU)_2_ and ISU–SU in Scheme 2), and the oxidation products (ISOU and IU in Scheme 2). The last two products involved H_2_O_2_ (a well-known water radiolysis product resulting from recombination of ^●^OH radicals) [20], while the dimers utilized the ^●^*t*-BuO radical (abundant under the experimental conditions produced in the reaction between the ^●^OH radicals and *t*-BuOH molecules). All the reactions discussed are shown in Scheme 2.

#### 2.2.1. Products Induced by Solvated Electrons

One of the expected products of ISU irradiation, SU, results from the DEA, according to the mechanism that was proposed previously [3,24,25]. As shown in Scheme 2, the electron e^−^ produced by water radiolysis attaches to the neutral molecule of ISU, forming the anion radical, which (via a transition state) is readily dissociated to the I^−^ anion and the C5-centered SU^●^ radical. Next, the radical abstracts a hydrogen atom from the environment, forming the neutral product SU. This process is both thermodynamically favorable (for ISU decomposition to I^−^ anion and SU^●^ radical ΔG = −80.8 kcal/mol) and associated with a very low kinetic barrier (ΔG* = 2.8 kcal/mol for the C5–I bond breakage), which suggests that ISdU is a potent radiosensitizer under hypoxic conditions (where e^−^_aq_ are as abundant as the ^●^OH radicals) [20].

#### 2.2.2. Oxidation Products

The oxidation products, ISOU and SU, usually result from the reaction with singlet oxygen formed via photochemical oxidation [26,27,28]. Yet, singlet oxygen does not form during irradiation with IR, as was indicated by the X-ray studies on silicon nanoparticles involving a fluorescence probe for singlet oxygen [29]. Therefore, we suggest that hydrogen peroxide, a product of the recombination of two ^●^OH radicals [20], is responsible for both products, according to the desulfurization mechanism of thiocarbonyl compounds proposed previously [30]. This hypothesis is additionally supported by the fact that in additional experiments, conducted without the ^●^OH scavenger, we were able to observe a higher yield of ISOdU (Appendix A). It is also worth noting that H_2_O_2_ is widely recognized as the oxidative agent for thiols and sulfides [31,32,33,34] and has been shown to oxidize thioamides, which is the case in the current study [30].

It was shown previously that the reactions mentioned above are highly affected by explicit waters included in the computational model, which questions the usage of the polarizable continuum model (PCM) approach [31]. These explicit waters provide essential stabilization for charge separation and significantly lower activation barriers. Therefore, we performed calculations for the oxidation of ISU to ISOU using two computational models, including only reactants, and with three explicit water molecules, as suggested by Chu and Trout [31]. In the first case, the kinetic barrier and thermodynamic stimuli were equal to 25.5 kcal/mol and −30.6 kcal/mol, respectively, while with explicit waters, they were 15.3 kcal/mol and −43.2 kcal/mol, respectively (for both transition state structures see Appendix A). This trend is consistent with that presented previously for dimethyl sulfide [31], and our values for the three explicit waters seem to justify the formation of ISOU in the current experiments.

On the other hand, the IU product may be formed from ISOU via a cyclic intermediate (oxathiirane) that undergoes sulfur extrusion [30,35]. Using the same model as before (with explicitly added water molecules), we obtained the barrier for unstable oxathiirane formation equal to 24.5 kcal/mol and the subsequent barrier-free opening to IOSU with an overall thermodynamic stimuli of −4.9 kcal/mol, as shown in Scheme 2. This product was then ready to extrude the sulfur, giving IU in the final step, as suggested in the literature (Scheme 2) [36,37,38,39,40].

#### 2.2.3. Formation of Dimers

In contrast to products formed via pathways B and C (Scheme 1), the dimer yield decreases if oxygen is present in the system (Appendix A). Moreover, it was additionally observed that if *t*-BuOH is absent, the dimer yield dramatically falls, which suggests the involvement of *t*-BuO^●^ radicals in the studied reactions (Appendix A). Hence, the proposed pathway of dimer formation is based on the previous studies on 4-thiouridine disulfide [41,42] supplemented with the current observations.

The dimer, formed from two identical ISU fragments, was calculated to simply be the product of recombination of two S-centered ISU^●^ radicals after they were formed as a result of the reaction of *t*-BuO^●^ radicals with ISU. For the more complicated dimer, ISU–SU, the mechanism presented in Scheme 2 involved the DEA process where the SU^●^ radical was formed as discussed previously (see Section 2.2.1), followed by the N(3)H to C(5)H shift, as suggested by Wenska et al. [42]. The two S-centered radicals, SU^●^ and ISU^●^, may then react forming the observed ISU–SU dimer.

### 2.3. Clonogenic Assay

Clonogenic assays were used to assess the survival and proliferation potential of MCF-7 human breast cancer cells treated with ISdU and/or IR. The studied derivative enhances the sensitivity of MCF-7 cells to radiation (for an image of colonies formed after ISdU and/or X-ray treatment see Appendix A). The radiosensitizing effect of ISdU (see Figure 2) depends on its concentration (0, 10, or 100 µM) and radiation dose (0, 0.5, 1, 2, or 3 Gy). In the tested cell line, the clonogenic assay revealed that pre-treatment with ISdU reduced the survival of the cells irradiated with 0.5 Gy from 78.4 ± 0.9% to 67.7 ± 0.3% after incubation with 10 µM ISdU solution and to 59.8 ± 1.9% after incubation with 100 µM ISdU. With a slightly higher dose (1 Gy), the surviving fraction was reduced from 68.2 ± 0.8% to 54.9 ± 0.2% after incubation with 10 µM ISdU solution and to 40.8 ± 1.4% after incubation with 100µM ISdU solution. Also, for higher doses of IR (2 Gy and 3 Gy), ISdU caused a significant reduction in the proliferation of cancer cells. Low doses of radiation, at which the excellent sensitizing effect of ISdU is observed, suggest that the studied compound is a promising radiosensitizer.

### 2.4. MTT Assay

The concept of using ISdU as a sensitizer in radiotherapy requires that it should be marginally cytotoxic in the absence of IR. For this reason, the cytotoxicity of ISdU towards (i) human breast cancer cells from the MCF-7 line and (ii) human dermal fibroblasts (HDFa line) was examined using the MTT assay. It was tested in six concentrations (in the range of 0–10^−4^ M) and two time variants (24 h and 48 h incubation). Figure 3a demonstrates that ISdU caused a statistically significant reduction in MCF-7 cells viability compared to the control (up to 95 ± 3% for 24 h incubation and up to 90 ± 3% for 48 h incubation) only for the highest concentration tested (10^−4^ M). In other cases, a decrease in cell viability of the MCF-7 line was not observed. The MTT assay also showed a negligible cytotoxic effect of ISdU toward normal human dermal fibroblasts (Figure 3b). The reduction in cell viability at all tested concentrations was not statistically significant. The obtained results allow us to consider ISdU as a compound with very low cytotoxicity towards both normal and cancer cells, and this fact opens the route to studies on animal models.

### 2.5. Histone H2A.X Phosphorylation and Cell Death Assays

Histone H2A.X phosphorylation was used as a marker of DNA double-strand breaks (DSBs) [43]. To determine the relationship between γH2A.X formation and radiosensitivity, cytometric analysis was performed. γH2A.X formation was detected 1 h after irradiation with doses of 0 Gy, 1 Gy, and 2 Gy for samples treated with the studied compound (48 h incubation with 10^−4^ M ISdU) and non-treated cultures. Our studies demonstrate that treatment with ISdU results in an increase in the population of γH2A.X positive cells after irradiation with the doses of 1 and 2 Gy (Figure 4 and Appendix A). After incubation with ISdU and irradiation with a dose equal to 1 Gy, the level of γH2A.X increased from 26.56 ± 1.14% (for the non-treated control) to 35.15 ± 1.48%. The combination of ISdU treatment and 2 Gy of ionizing radiation resulted in the enhancement of the γH2A.X fraction from 49.38 ± 3.8% to 60.25 ± 2.98%. These results suggest that ISdU sensitizes MCF-7 cells to ionizing radiation, at least in part, by increasing the formation of DSBs.

In addition, we performed a cytometric assay which allowed multiparametric evaluation of three cell health markers: change in mitochondrial potential (early apoptosis and cellular stress), phosphatidylserine expression on the cell surface (late apoptosis), and membrane permeabilization (cell death). Our results (Appendix A) confirm that pretreatment with ISdU leads to increased radiosensitivity of MCF-7 cells (reduction of cell viability and increase in the population of cells in early apoptosis). Indeed, irradiation with a dose of 2 Gy caused reduction in cell viability from 87.5 to 81.17% for non-treated cultures and from 88.36 to 75.35% for cultures treated with ISdU (see Appendix A). The population of dead cells increased after irradiation from 11.13% (for the control culture) to 18.48% (for the ISdU treated culture; see Appendix A). To our surprise, the population of apoptotic cells did not change with irradiation and exposure to ISdU. It is, however, worth noticing that cell population in early apoptosis increased as a result of both irradiation and pretreatment with ISdU (see Appendix A). The negligible influence of irradiation and ISdU on the population of apoptotic cells can be comprehended by that fact that the MCF-7 breast carcinoma cells are resistance to the apoptosis induced by ionizing radiation, which is a consequence of their lack of caspase-3 [44]. 

## 3. Materials and Methods 

### 3.1. Experimental Methods 

#### 3.1.1. Synthesis of 3′,5′-di-*O*-acetyl-2′-Deoxyuridine

A solution of 2′-deoxyuridine (1.00 g, 4.38 mmol) in pyridine (12 mL) was stirred at room temperature with acetic anhydride (911 µL, 9.64 mmol) for 24 h. The syrupy residue was co-evaporated with three portions of aqueous ethanol (10 mL) and *n*-heptane to remove the pyridine residues. The raw di-*O*-acetyl derivative was obtained quantitatively.

#### 3.1.2. Synthesis of 3′,5′-di-*O*-acetyl-5-iodo-2′-Deoxyuridine

A mixture of 3′,5′-di-*O*-acetyl-2′-deoxyuridine (1.37 g, 4.39 mmol), iodine (0.667 g, 2.63 mmol), and ceric ammonium nitrate (CAN) (1.21 g, 2.20 mmol) in ACN (70 mL) was stirred at 80 °C for 1 h. After that time, the solvent was evaporated, and the residue was treated with 5% NaHSO_3_/H_2_O (10 mL) to neutralize the unreacted iodine. After that, a mixture was extracted with EtOAc and washed with brine. The combined organic layers were washed with water, dried with MgSO_4_, and evaporated. The crude 5-iodo-3′,5′-di-*O*-acetyl-2′-deoxyuridine was crystallized from ethanol to give white crystals of purified product (1.28 g, 66.7%).

#### 3.1.3. Synthesis of 3′,5′-di-*O*-acetyl-5-Iodo-4-Thio-2′-Deoxyuridine

3′,5′-di-*O*-acetyl-5-iodo-2′-deoxyuridine (1.28 g, 2.92 mmol) was dissolved in 1,4-dioxane (40 mL) and P_2_S_5_ (1.95 g, 8.76 mmol) was added. The mixture was refluxed until thin layer chromatography (TLC) analysis (CHCl_3_:CH_3_OH, 30:1) showed complete disappearance of the substrate (2–3 h). The solvent was removed under reduced pressure and the residue was treated several times with CHCl_3_. The combined chloroform extracts were evaporated, and the residue was separated on a silica gel column, which was eluted with CHCl_3_:CH_3_OH, 20:1. After evaporation, the desired product was obtained as a yellow solid (1.06 g, 80.3%).

#### 3.1.4. Synthesis of 5-Iodo-4-Thio-2′-Deoxyuridine

3′,5′-di-*O*-acetyl-5-iodo-4-thio-2′-deoxyuridine (1.06 g, 2.33 mmol) was dissolved in methanol (25 mL) and stirred at 0 °C. A methanolic sodium methoxide solution (0.186 g, 4.66 mmol), freshly prepared from NaH and anhydrous methanol, was added in portions. The mixture was stirred at room temperature until TLC analysis showed complete disappearance of the substrate (10 min). The mixture was purified on a silica gel column, which was eluted with CHCl_3_:CH_3_OH, 20:1. The final product was obtained as a yellow solid (0.428 g, 47.8%).

^1^H NMR (Bruker AVANCE III, 500 MHz, DMSO), δ: 12.99 (s, 1H), 8.56 (s, 1H), 6.01 (t, 1H), 5.27 (s, 1H), 5.20 (s, 1H), 4.20–4.26 (m, 1H), 3.79–3.83 (m, 1H), 3.61–3.67 (m, 1H), 3.52–3.59 (m, 1H), and 2.13–2.21 (m, 2H); ^13^C NMR (125 MHz, DMSO), δ: 189.8, 148.1, 142.6, 88.2, 85.8, 83.5, 70.0, 60.9, and 40.8. HRMS (TripleTOF 5600+, SCIEX), m/z: [M − H]^−^ calculated for C_9_H_10_IN_2_O_4_S 369.1567 found 368.9564; UV spectrum (water), λ_max_: 350 nm. For the synthesis scheme, NMR, and mass spectra see Appendix A.

#### 3.1.5. Radiolysis

Radiolysis of ISdU solution (10^−4^ M) containing 0.03 M *t*-BuOH as a scavenger of the ^●^OH radicals and phosphate buffer (10 mM, pH = 7.0) was carried out in a Cellrad X-ray cabinet (Faxitron X-ray Corporation, Tucson, AZ, USA). All samples were deoxygenated by purging with argon for ~3 min and exposed to 140 Gy. The studied samples were analyzed in triplicate.

#### 3.1.6. HPLC Analysis

A reversed-phase HPLC method was employed for analysis of the irradiated and non-irradiated samples of ISdU (Dionex UltiMate 3000 System with a diode array detector, Waltham, MA, USA), which was set at 260 nm for monitoring the effluents. Separation by HPLC was achieved using a C18 column (Wakopak Handy ODS, 4.6 × 150 mm, 5 μm in particle size and 100 Å in pore size) with a gradient elution of 80% acetonitrile and 0.1% formic acid (from 0 to 40% acetonitrile) and flow rate 1 mL·min^−1^.

#### 3.1.7. LC–MS Analysis

ISdU and radiolysis products were directly analyzed by LC–MS and LC–MS/MS. The LC–MS and LC–MS/MS experiments were performed using a TripleTOF 5600+ (SCIEX) mass spectrometer (operated in negative mode) coupled with the ultra high performance liquid chromatography (UHPLC) system Nexera × 2. Chromatographic conditions were a Kinetex column (Phenomenex, 1.7 µm, C18, 100 Å, 2.1 × 150 mm), a flow rate of 0.2 mL·min^−1^, and a gradient elution with 80% acetonitrile and 0.1% formic acid (from 0% to 40% acetonitrile). The column temperature was maintained at 25 °C. The effluent was diverted to waste for 2 min after injection during each analysis. Conditions for MS and MS/MS analysis were a spray voltage of −4.5 kV, nebulizer gas (N_2_) pressure of 25 psi, flow rate of 11 L·min^−1^, and a source temperature of 300 °C. Each spectrum was obtained by averaging three scans, and the time of each scan was 0.25 s.

#### 3.1.8. Clonogenic Assay

Cells (MCF-7 human breast cancer cell line obtained from Cell Lines Service (CLS), Eppelheim, Germany) were plated at a density of 10^6^ cells per 60 mm dish. The cells were treated with ISdU in concentrations of 10^−4^ and 10^−5^ M. After 48 h of treatment, the cells were trypsynized, plated at a density of 800 cells per 100 mm dish and incubated under 37 °C, 5% CO_2_. The cells were grown in RPMI medium (Roswell Park Memorial Institute medium) supplemented with 10% fetal bovine serum (FBS) and antibiotics (streptomycin and penicillin) at a concentration of 100 U·mL^−1^. After 16 days the resulting colonies were fixed with 6.0% (*v*/*v*) glutaraldehyde and 0.5% crystal violet. Stained colonies were counted manually, and colony size was assessed using an inverted fluorescence microscope (Olympus, IX73). The experiment was carried out in duplicate. Non-irradiated cultures were used as controls. Plating efficiencies are shown in Appendix A.

#### 3.1.9. Cytotoxicity Assay

Cytotoxic activity was determined using the MTT assay [45] after incubation of MCF-7 or HDFa (obtained from Gibco) cells with ISdU. The MCF-7 cells were grown in RPMI supplemented with 10% FBS and antibiotics (streptomycin and penicillin) at a concentration of 100 U·mL^−1^. The HDFa cells were grown in DMEM (Dulbecco’s Modified Eagle medium) supplemented with 10% FBS and antibiotics (streptomycin and penicillin) at the same concentration. Cells at a density of 4·10^3^ per well were seeded into a 96–well plate and incubated (37 °C, 5% CO_2_) overnight. Then, the medium was freshly changed, and the cells were treated with ISdU at concentrations of 0 (control), 10^−4^, 5 × 10^−5^, 10^−5^, 10^−6^, 10^−7^, and 10^−8^ M. Cells were incubated with ISdU for 24 h and 48 h. After this time, the aqueous MTT salt solution at a concentration of 4 mg·mL^−1^ was added to each well. After 4 h of incubation, the medium was removed and 200 μL of dimethylsulfoxide (DMSO) was added to dissolve the formazan crystals. The absorbance was measured at 570 nm (with 660 nm taken as the reference) on an EnSpire microplate reader (PerkinElmer). The viability of controls was taken as 100%. The experiment was carried out in at least four replications. The results were analyzed with the use of GraphPad Prism software. The statistical evaluation of treated samples and untreated controls was calculated using one-way analysis of variance (ANOVA) followed by Dunnett’s multiple comparison test. The data was obtained from three independent experiments and each treatment condition assayed in triplicate. The differences were considered significant at *p* < 0.05.

#### 3.1.10. Histone H2A.X Phosphorylation Assay

MCF-7 cells were grown in RPMI supplemented with 10% FBS and antibiotics at a concentration of 100 U·mL^−1^. Cells were treated with ISdU at a concentration of 10^−4^ M and incubated (37 °C, 5% CO_2_) for 48 h. After this time, the plates with cells were irradiated (Cellrad X-ray cabinet, Faxitron X-ray Corporation) with 1 Gy and 2 Gy doses and incubated for 1 h. Then, the cells were dissociated with Accutase solution, fixed, and permeabilized. Next, the cells were stained and analyzed by flow cytometry (Guava easyCyte™ 12, Merck). Fixation, permeabilization, and staining were carried out according to the manufacturer’s protocol (FlowCellect™ Histone H2A.X Phosphorylation Assay Kit, Merck). The experiment was carried out in triplicate. Non-treated cultures were used as controls.

#### 3.1.11. Cell Death Assay

Cells were treated with ISdU at a concentration of 10^−4^ M and incubated (37 °C, 5% CO_2_) for 48 h. After this time, the plates with cells were irradiated (Cellrad X-ray cabinet, Faxitron X-ray Corporation) with a dose of 2 Gy and incubated for 1 h. Then, the cells were dissociated with Accutase solution, stained, and analyzed by flow cytometry (Guava easyCyte™) according to the manufacturer’s protocol (FlowCellect™ MitoDamage Kit, Merck).

### 3.2. Computational

All calculations were performed with the use of the B3LYP functional [46,47,48] and DGDZVP++ basis sets. The DGDZVP++ acronym stands for the DGDZVP basis set augmented with one extra set of sp symmetry diffuse functions added to hydrogen and heavy atoms [49,50]. For every symmetry, the exponent of diffuse basis function was equal to 1/3 of the smallest exponent in the original DGDZVP basis set [51]. The water environment was simulated with the use of the PCM [52,53]. In some cases, explicit water molecules were required, as discussed in the results section (for the structure with explicit water molecules, we adopted the relative positions of water and sulfur from appropriate structures presented by Chu et al. (TS-7) [31], added the rest of the molecule, and reoptimized). The calculations were performed for the uracil derivative substituted with the methyl group at the N1 position (5-iodo-1-methyl-4-thiouracil, ISU, Scheme 2) to mimic the sugar-binding side present in the nucleoside. All calculations were carried out with the Gaussian 09 package [54].

## 4. Conclusions

In this paper, we suggest that ISdU, besides having known photosensitizing properties, also possesses radiosensitizing features as well as demonstrate that it effectively sensitizes breast cancer cells (MCF-7 line) to X-rays. ISdU belongs to the group of modified nucleosides, which undergoes DEA induced by solvated electrons (one of the major products of water radiolysis under hypoxic conditions, characteristic for solid tumors) that are negligibly reactive towards native DNA. In order to verify the propensity of ISdU to be damaged by solvated electrons, we carried out radiolysis of deoxygenated ISdU solution in the presence of an ^●^OH radical scavenger and phosphate buffer. It turned out that the radiolysis of the studied derivative, caused by X-ray irradiation, results in the generation of five major radioproducts. The pathways for the formation of these radioproducts were calculated quantum-chemically with the use of the density functional theory. One of the major products of ISdU irradiation, SdU, results from the DEA process. Besides ISOdU and IdU (oxidation products), the (ISdU)_2_ and ISdU–SdU dimers were identified. The decomposition of ISdU is almost 1.5-fold more effective compared to BrdU, a well-known radiosensitizer. In order to get deeper insight into the mechanism, pulse radiolysis experiments are currently being performed.

With such favorable radiolytic characteristics, we decided to carry out a clonogenic assay, which proved that the treatment of breast cancer cell cultures with ISdU resulted in a significant decrease (about 20%) of their survival after irradiation, even with doses as low as 0.5 Gy. A histone H2A.X phosphorylation assay showed that ISdU sensitizes MCF-7 cells to ionizing radiation, at least in part, by increasing the formation of DSBs while the MitoDamage cell death assay confirmed that pretreatment of MCF-7 cells with ISdU leads to the IR-induced reduction of cell viability and an increase in the population of cells in early apoptosis. The studied derivative turned out to be non-cytotoxic towards both normal and cancer cells; therefore, only after irradiation are the lethal effects observed inside the cell. The present studies established ISdU as a potential radiosensitizer that after incorporation into DNA and exposure to IR should cause lethal damage, such as strand breaks, both intra- and interstrand DNA crosslinks, as well as DNA–protein crosslinks. Use of effective radiosensitizers results in an increase of the efficacy of radiotherapy, allowing the reduction of the employed radiation dose, which eventually gives radiological protection to healthy tissues. The biological studies of the intracellular metabolism of ISdU are currently underway in our laboratory.

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
