# Peer review of "5-Iodo-4-thio-2′-Deoxyuridine as a Sensitizer of X-ray Induced Cancer Cell Killing"

_ijms, 2019, doi:10.3390/ijms20061308_

Round 1
Reviewer 1 Report
The manuscript is much improved, especially with the H2AX data.
Reviewer 2 Report
The authors have carefully addressed the concerns and comments raised.
This manuscript is a resubmission of an earlier submission. The following is a list of the peer review reports and author responses from that submission.
Round 1
Reviewer 1 Report
ISdU has previously been shown to induce cell killing after UVA irradiation. The present study investigates the potetial of ISdU as a sensitizer of ionizing radiation and propose a molecular mechanism for the sensitization based on steady state analysis of the radiolysis products.
The manuscript is well written and the study seems to be well performed.
Minor issue: The plating efficiencies for the controls with and without pretreatment should be included.
Author Response
We would like to
thank the Referees for their valuable comments which helped us to
improve the quality of the current work. Our detailed response and the
list of corrections addressing all their remarks are listed below.
Reviewer 1: 1.“Minor issue: The plating efficiencies for the controls with and without pretreatment should be included.”
Indeed, we did not specify the plating efficiency. This information has been included in the revised version of the manuscript: “The average plating efficiencies for the controls with and without pretreatment are equal to 26.31% (0 µM), 23.81% (10 µM) and 21.31% (100 µM).” – see page 7, lines 217-218.
Additionally, plating efficiencies for all culture variants are shown in Table S1 in Supplementary Materials.
Reviewer 2 Report
The specific study targets the role of 5‐iodo‐4‐thio‐2’‐deoxyuridine (ISdU) as a compound that can effectively lead to ionizing radiation (IR)‐induced cellular death in other words acting as a radiosensitizer even at relatively low doses of 0.5-1 Gy. Although the chemical work (characterization of radiation products, HPLC etc.) stands very well, the biological and mechanistic input is problematic. The cytotoxicity assay is ok but with the use of only one cell line (malignant) without any normal one, not specific assay for DNA damage assesment (like gamma-H2AX or Comet assay for oxidative bases, single strand breaks or DSBs), the conclusios are weak. In addition, there is not any specific assay for cell death (clonogenic survival is not only for cell death) and the problems are augmented with a discussion and introduction partially correct and complete without for example mentioning the role of complex DNA damage (DSBs and non-DSB lesions-OCDLs) that maybe induced at these doses (2013: Induction and repair of clustered DNA Lesions: What do we know so far? Radiat. Res., 180, 100–109). The English and language must be proofread by a native English speaker.
Author Response
Please see authors' responses to reviewer 2 in the attachment.
